# The "ready-to-hand" test: Diagnostic availability and usability in primary health care settings in Sierra Leone

Alice Street [1‡]*, Eva Vernooij [1], Francess Koker[2], Mats Stage Baxter [3], Fatmata Bah[2], James Rogers[4], Momoh Gbetuwa[5], Mikashmi Kohli [6], Rashid Ansumana[7‡]

**1** Department of Social Anthropology, School of Social and Political Sciences, University of Edinburgh, Edinburgh, United Kingdom, **2** Kings Sierra Leone Partnership, King's Centre for Global Health and Health Partnerships, Freetown, Sierra Leone, **3** Usher Institute, University of Edinburgh, Edinburgh, United Kingdom, **4** James Rogers, Laboratory Technical Working Group, Sierra Leone Ministry of Health and Sanitation, Freetown, Sierra Leone, **5** Momoh Gbetuwa, College of Medicine and Allied Health Sciences, University of Sierra Leone, Freetown, Sierra Leone, **6** Mikashmi Kohli, FIND, Campus Biotech, Geneva, Switzerland, **7** Rashid Ansumana, School of Community Health Sciences, Njala University, Bo, Sierra Leone

‡ AS and RA are joint senior authors on this work.
* alice.street@ed.ac.uk

**Data Availability Statement:** The anonymised data on which this study is based can be accessed at https://doi.org/10.7488/ds/3401.

## Abstract

This article assesses the availability of essential diagnostic tests in primary health care facilities in two districts in Sierra Leone. In addition to evaluating whether a test is physically present at a facility, it extends the concept of availability to include whether equipment is functional and whether infrastructure, systems, personnel and resources are in place to allow a particular test to be "ready to hand", that is, available for immediate use when needed. Between February 2019 and September 2019, a cross-sectional mixed-methods survey was conducted in all 40 Community Health Centres (CHCs) in Western Area, one of five principal divisions in Sierra Leone. The number of rapid diagnostic tests (RDTs) available ranged from 1–12, with 75% of facilities having 9 or less RDTs available out of a possible 17. While RDTs were overall more widely present than manual assays, there was wide variation between tests. The presence of RDTs at individual facilities was associated with having a permanent laboratory technician on staff. Despite CHCs being formally designated as providing laboratory services, no CHC fulfilled standard World Health Organisation (WHO) criteria for a laboratory. Only 9/40 (22.5%) CHCs had a designated laboratory space and a permanently employed laboratory technician. There was low availability of essential equipment and infrastructure. Supply chains were fragmented and unreliable, including a high dependency (>50%) on informal private sources for the majority of the available RDTs, consumables, and reagents. We conclude that the readiness of diagnostic services, including RDTs, depends on the presence and functionality of essential infrastructure, human resources, equipment and systems and that RDTs are not on their own a solution to infrastructural failings. Efforts to strengthen laboratory systems at the primary care level should take a holistic approach and focus on whether tests are "ready-to-hand" in addition to whether they are physically present.

**Funding:** This research was supported by the European Research Council under EU Horizon 2020 programme (Grant Number 715450, awarded to AS). The funders had no role in study design, data collection and analysis, decision to publish, or preparation of the manuscript.

**Competing interests:** The authors have declared that no competing interests exist.

## Introduction

### Background

The lack of perceived urgency around diagnostic testing in the global health community has had profound implications for resources and investments: the recently published 2021 Lancet Commission on Diagnostics found that 47% of the global population has little or no access to diagnostics [1]. Yet, awareness of the importance of diagnostic testing in under-resourced settings is growing alongside concerns about anti-microbial resistance, rising healthcare costs, and the threat of emerging pathogens [2]. The development of a new generation of affordable, easy to use, and portable diagnostic devices is also radically transforming access to diagnostics in settings without laboratory infrastructure or expertise [3, 4]. One sign of progress was the publication of the first Essential Diagnostic List (EDL) by the WHO in 2018, nearly 45 years after the first essential medicines list was released. The EDL identifies the basic *in vitro* diagnostics that should be available at the point of care and in laboratories, with the aim of assisting governments to develop their own national lists and providing guidance to developers on diagnostic demand [5, 6].

Since the publication of the EDL, a small number of studies have used it as the basis for research to assess the availability of essential tests at the primary care level in different countries including India [7], Peru [8], Ghana [9] and Nigeria [10]. These studies have shown that the availability of essential tests is frequently poor, with some EDL tests universally unavailable in some countries, and large discrepancies of availability of tests between different health facilities and across different regions or districts. In this article, we build on this growing evidence base to assess the availability of essential diagnostic tests and diagnostic systems in primary health care facilities across the Western Area in Sierra Leone. Conducted five years after the West Africa Ebola Virus Disease (EVD) outbreak, which drew international attention to the under-resourcing of national diagnostic systems in the region [11, 12], our study seeks to identify outstanding resource gaps in the provision of essential diagnostic services at a primary care level.

We expand the meaning of availability beyond a narrow definition of physical presence to include whether or not tests are "ready to hand" for the people, in this case laboratory and healthworkers, who use them. By "ready to hand" we refer to the immediate handiness, usefulness and meaningfulness of testing equipment and resources to staff, without the need for them to undertake further cognitive or physical effort to make them useable. This approach recognises that tests are practices that people *do* rather than discrete, self-contained things that can be counted [13–15]. Manual laboratory assays, for example, are testing practices that comprise an extensive assemblage of equipment, reagents, standard operating procedures, and laboratory staff. Even rapid diagnostic tests (RDTs), which are expressly designed to operate independently of laboratory infrastructure, often come as a testing "kit" made up of several different elements and still need to be ordered, stored, used and disposed of by appropriately trained personnel [16, 17]. We therefore assess the availability and functioning of human resources, physical infrastructure and systems and examine their impact on the ability of health facility staff to provide testing services.

In focusing on what makes tests "ready-to-hand" we aim to contribute to academic and policy discussions about the future direction of laboratory strengthening in LMICs. We question the common opposition of infrastructure-heavy laboratory testing to infrastructure-light RDTs–a distinction that is also implicitly made by the EDL's distinction between tests that should be available at facilities with a laboratory and those that should be available at facilities without a laboratory. Instead we show that laboratories can take many forms and that the availability of RDTs can also depend on laboratory infrastructure. We end the paper with a

discussion of its implications for the EDL and for the development of a national EDL in Sierra Leone.

## Methods

### Setting

Sierra Leone's Basic Package of Essential Health Services (BPEHS, from here on referred to as the Basic Package), first published in 2010 and most recently updated in 2015, outlines the infrastructural needs of laboratories and diagnostic tests at different levels of the country's health system [18, 19]. It stipulates that basic laboratory services should be available at the Community Health Centre (CHC) level, with more specialised laboratory testing, imaging services, and blood services provided in District and Regional Hospitals. CHCs are situated at the Chiefdom level and serve a population of between 10,000 to 30,000. They are staffed by a Community Health Officer, Community Health Nurses, Midwives, Laboratory Technicians, Laboratory Assistants and environmental health workers. In Sierra Leone's tiered health system, CHCs sit above Maternal and Child Health Posts (MCHPs) and Community Health Posts (CHPs), both of which are staffed by community health workers and which are not required by the Basic Package to offer laboratory services.

Data collection was undertaken in Western Area, one of five principal administrative subdivisions in Sierra Leone. The area includes two districts; Urban and Rural, which with a population of 1.5 million [20] are the most densely populated districts in Sierra Leone. The country's five referral hospitals are all located in Western Area, three in Urban District and two in Rural District. The country's two national referral laboratories are located in Western Area Rural District. At the time of data collection, Western Area had 4 district government hospitals, 40 CHCs and 28 CHPs [21].

### Data sampling and collection

All CHCs in Western Area Rural District ($n = 15$) and Urban District ($n = 25$) were included in the survey, totalling 40 CHCs. Between February 2019 and September 2019, a researcher (FK) visited the 40 CHCs in Western Area to conduct a check-list style cross-sectional survey, completed in real-time with the assistance of facility staff.

### Survey tools and variables

The checklist format of the survey broadly followed that used by Kohli and colleagues in their survey of diagnostic availability in two districts in India [7]. At the request of the Sierra Leone Ministry of Health and Sanitation (MOHS), the survey was based on the Basic Package as the national benchmark for essential tests that match the disease profile and meet the needs and priorities of the country. The Basic Package included 27 tests for the CHC level, 16 of which were RDTs in the form of lateral flow dipsticks, cassettes or handheld devices. Most, but not all of the tests listed in the EDL were included in the Basic Package (see S1 Text). In addition to the tests listed in the Basic Package, the survey included two tests (the manual Widal test and the Cholera RDT), which were of interest to the MOHS because of their perceived high popularity and availability in Sierra Leone, despite having limited sensitivity.

During visits to the CHCs the researcher requested to view the RDTs and laboratory infrastructure to verify its availability. To provide additional detail on whether a diagnostic device that is reported as available is actually "ready-to-hand" for laboratory workers, we did observational checks of the equipment, consumables and reagents needed to carry out the five manual assays with the highest reported availability. In addition, the survey included questions about

supply sources for tests and reagents, staffing, and availability of essential infrastructure and systems (electrification, water supply, maintenance, biosafety, and waste management) (see S1 Text).

The survey tool also included qualitative questions to establish contextual reasons for the presence or absence of particular items on the checklist. In-depth ethnographic research was carried out at one of the CHCs included in the survey sample, including the structured observation of patients' diagnostic pathways, observation of laboratory practices, and interviews with facility staff [21, 22]. Contextual research into the history of laboratory strengthening in Sierra Leone and policy changes implemented since the end of the 2014–2016 EVD outbreak included review of grey literature and policy documents, 14 interviews with key informants in the MOHS and international organisations involved in laboratory strengthening initiatives, and observational site visits to 12 clinical and reference laboratories [22].

## Data analysis

Hard-copies of the survey forms were scanned and the data manually copied into Microsoft Excel 16.43 and imported into SPSS 25.0 for analysis. Descriptive statistical analyses were conducted to assess the availability of diagnostic tests, reagents, consumables, equipment, CHC general characteristics, facility infrastructure, human resources, waste management and disposal procedures. Chi-square test and Fisher's exact test were used to compare availability of diagnostic tests between 1) rural and urban districts and 2) whether or not a permanent lab technician was available on-site. Fisher's exact test was used if expected frequencies had a value of <5. A significance level (alpha) of 5% was adopted. Cramér's V was used to determine the strength of association between the categorical variables mentioned above. The interpretation of the effect size by Rea and Parker was used [23]. Thematic analysis of qualitative responses to the survey and ethnographic research data was undertaken to identify common causes of unavailability, including whether those causes pertained to facility-specific factors or broader health system issues. Missing data was omitted from both descriptive and bivariate analyses, thus only utilising complete case analyses.

## Ethics

Ethical approval for the study was granted by the Sierra Leone Ethics and Scientific Review Committee and the University of Edinburgh Research Ethics and Integrity Committee. In addition, we obtained written approval from the Chief Medical Officer and Laboratory Director at the Sierra Leone Ministry of Health and Sanitation, and from two respective District Health Management Teams (DHMT) to carry out the study. Formal written consent was obtained from all interview participants. Formal verbal consent was obtained from all participants in the diagnostic availability survey. We do not present survey results for individual facilities in this publication or associated data in order to preserve anonymity for facility staff.

## Results

Our survey findings are structured in three sections. First we assess the diagnostic availability across both districts, presented separately for RDTs and assay-based tests. Second, we assess the association between test availability and human resources and key laboratory infrastructure. Third we assess the differences in dependence on supply systems between the two districts.

### Diagnostic availability

All 40 CHCs conducted some rapid diagnostic tests (RDT). Fig 1 shows the availability of RDT kits. Overall, facilities ranged from having between 1–12 different types of RDT kits available, with 75% of facilities having 9 or fewer out of the possible 17 RDT kits examined in the study. Malaria RDTs 97.5% ($n$/N = 31/40) and the HIV/Syphilis combined RDTs 77.5% ($n$/N = 31/40) were observed to be most available, which reflects findings from studies conducted in other countries [24]. The least available were testing kits for Leishmania and Haemoglobin glycated, neither of which were available in any of the 40 health centres. Five further testing kits were available in fewer than 25% of the facilities: Hepatitis B profile RDT, Hepatitis C test, the syphilis RDT, Helicobacter pylori RDT and Faecal occult RDT. The Hepatitis B RDT was the only RDT that showed significant difference in availability between rural and urban districts ($p$ = 0.024), with a moderate effect size suggesting that the RDT was more available in the rural district (Cramer's V = 0.388) (see Table A in S1 Data).

Fig 2 shows the availability of manual assays as reported by facility staff. Almost half of the facilities (19/40) had no manual assays available, while 75% of facilities had fewer than 5 manual assays available out of the 12 included in the study. The average availability percentage for all manual assays across all CHCs was 20.0%. By comparison, the average availability percentage for RDTs was 37.9%. The most available manual assay was the Widal test used to diagnose typhoid fever (37.5%, $n$/N = 15/40), despite not being listed in the Basic Package, followed by biological specimen routine analysis (35.0%, $n$/N = 14/40) and TB sputum microscopy (35.0%, $n$/N = 14/40). None of the manual assays showed statistically significant difference in availability between rural and urban CHCs (see Table B in S1 Data).

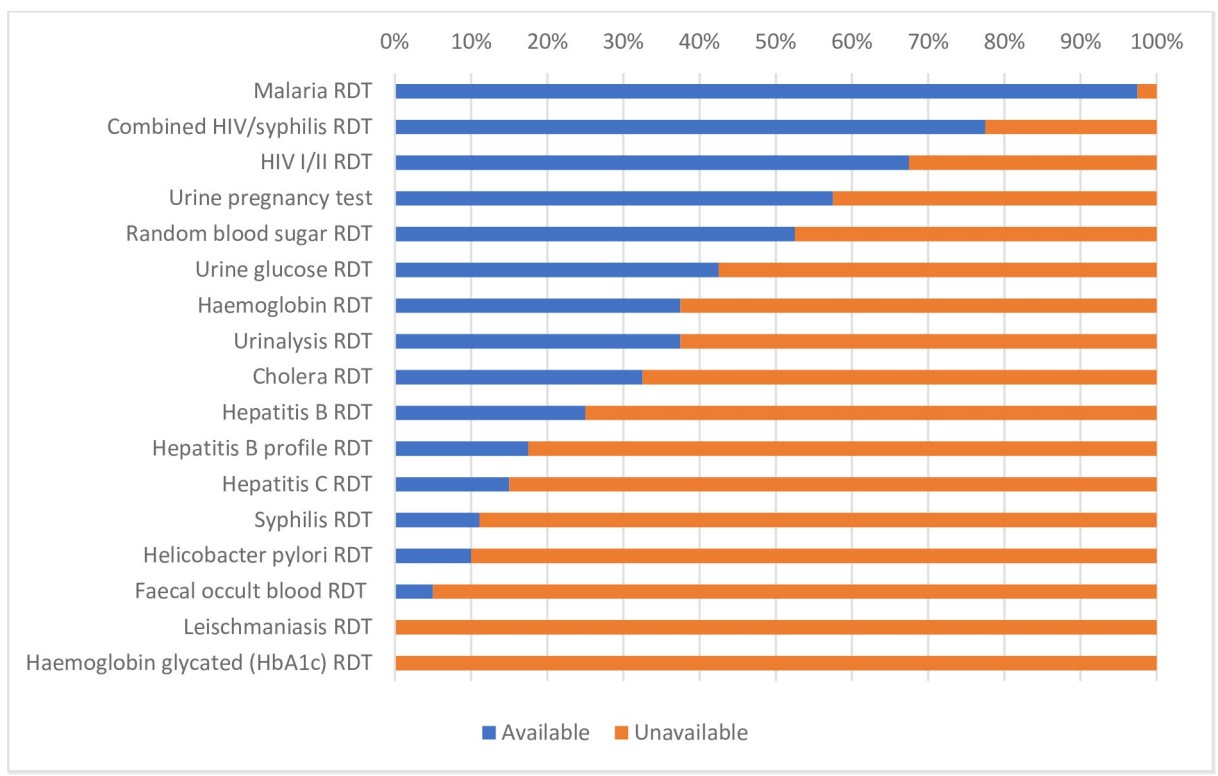

**Fig 1. Availability of RDTs.**

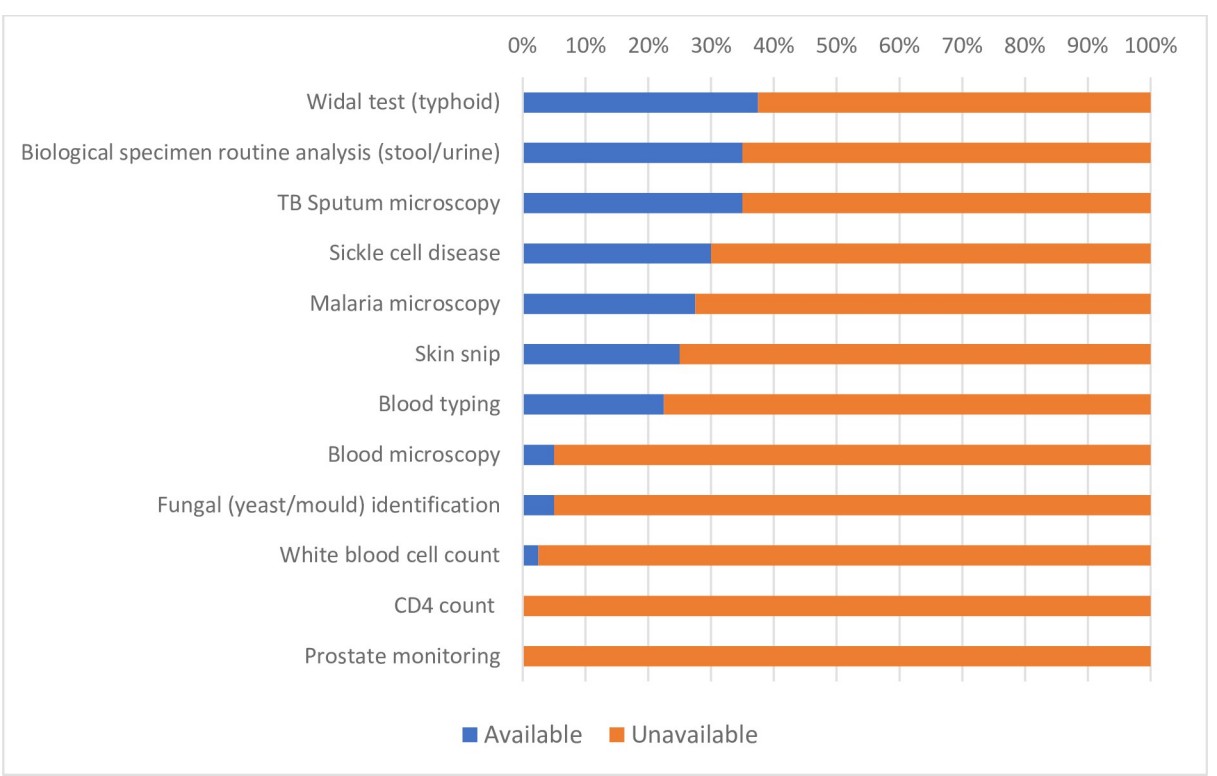

**Fig 2. Availability of assay-based tests.**

In Table 1 we present the availability of the equipment, consumables and reagents needed to carry out the five manual assays which had the highest reported availability (between 25.0% and 35.0%). We found multiple shortages in essential equipment and resources (see also Fig A in S1 Data). Less than half (42.5%, $n/N = 17/40$) of the CHCs had access to a working microscope and 70.0% ($n/N = 28/40$) of CHCs had a working refrigerator on-site. The lack of refrigerators (and cold chain provisions) affect the quality of diagnostic tests and reagents, and some staff reported that, in order to preserve quality, they stored diagnostic reagents in a fridge in one of the staff houses at the CHC compound. There were only two CHCs where the laboratory space had air conditioning, of which only one was working during the time of the data collection.

In some cases the observed availability of equipment, reagents and consumables necessary to carry out microscopy-based tests was lower than the reported availability of the test. For example, the availability of the Pasteur pipette (17.5%, $n/N = 7/40$), pipette stands (12.5%, $n/N = 5/40$), and spirit lamp (20.0%, $n/N = 8/40$) were lower than the reported availability of TB microscopy (35.0%, $n/N = 14/40$), but are all needed to carry out the test. For sickle cell disease which was reported to be available in 30.0% ($n/N = 12/40$) of the CHCs, the main reagent, sodium, was observed to be available in only 15.0% ($n/N = 6/40$) of the CHCs. Overall, the average availability percentage for all equipment, reagents and consumable items across all CHCs was 31.7% (equipment: 25.3%; consumables: 44.4%; reagents: 24.1%).

The differences in reported availability and observed availability of equipment necessary to conduct the tests revealed the level of improvisation that laboratory workers undertook to the make services available with the limited supplies they had. In some instances, for example, staff innovated on assay protocols with makeshift solutions, such as when the limited available of

**Table 1. Availability of equipment, consumables and reagents with the highest reported test availability (Malaria microscopy, TB sputum microscopy, biological specimen routine analysis (urine/stool microscopy), sickle cell disease, skin snip (oncho), and blood grouping).**

| Reagents, consumables, equipment | Total availability n/N (%) | Rural availability n/N (%) | Urban availability n/N (%) | Government supply n/N (%) | Government/private supply n/N (%) | Private supply n/N (%) |
|---|---|---|---|---|---|---|
| *Equipment* | | | | | | |
| Refrigerator | 28/40 (70.0%) | 10/15 (66.7%) | 18/25 (72.0%) | NA | NA | NA |
| Microscope | 17/40 (42.5%) | 8/15 (53.3%) | 9/25 (36.0%) | NA | NA | NA |
| Slide racks | 14/40 (35.0%) | 9/15 (60.0%) | 5/25 (20.0%) | NA | NA | NA |
| Spirit lamp | 8/40 (20.0%) | 2/15 (13.3%) | 6/25 (24.0%) | NA | NA | NA |
| Stopwatch | 7/40 (17.5%) | 5/15 (33.3%) | 2/25 (8.0%) | NA | NA | NA |
| Pasteur pipette | 7/40 (17.5%) | 3/15 (20.0%) | 4/25 (16.0%) | NA | NA | NA |
| Filtered pipette tips | 6/40 (15.0%) | 4/15 (26.7%) | 2/25 (8.0%) | NA | NA | NA |
| Incubator | 6/40 (15.0%) | 3/15 (20.0%) | 3/25 (12.0%) | NA | NA | NA |
| Pipette stands | 5/40 (12.5%) | 4/15 (26.7%) | 1/25 (4.0%) | NA | NA | NA |
| Water bath | 3/40 (7.5%) | 2/15 (13.3%) | 1/25 (4.0%) | NA | NA | NA |
| *Consumables* | | | | | | |
| Blood lancet | 35/40 (87.5%) | 12/15 (80.0%) | 23/25 (92.0%) | 30/35 (85.7%) | 1/35 (2.9%) | 4/35 (11.4%) |
| Cotton Wool | 28/40 (70.0%) | 12/15 (80.0%) | 16/25 (64.0%) | 18/28 (64.3%) | 1/28 (3.6%) | 9/28 (32.1%) |
| Needles and syringes | 28/40 (70.0%) | 11/15 (73.3%) | 17/25 (68.0%) | 14/26* (53.8%) | 0/26* (0.0%) | 12/26* (46.2%) |
| Nitrile disposable gloves | 26/40 (65.0%) | 13/15 (86.7%) | 13/25 (52.0%) | 18/26 (69.2%) | 0/26 (0.0%) | 8/26 (30.8%) |
| Microscopy slides | 19/40 (47.5%) | 9/15 (60.0%) | 10/25 (40.0%) | 15/19 (78.9%) | 0/19 (0.0%) | 3/19 (21.1%) |
| Blood collection tubes | 15/40 (37.5%) | 8/15 (53.3%) | 7/25 (28.0%) | 5/15 (33.3%) | 1/15 (6.7%) | 9/15 (60.0%) |
| Tourniquet | 10 /40 (25.0%) | 8/15 (53.3%) | 2/25 (8.0%) | 5/10 (50.0%) | 0/10 (0.0%) | 5/10 (50.0%) |
| Urine/stool container | 10/40 (25.0%) | 7/15 (46.7%) | 3/25 (12.0%) | 5/8* (62.5%) | 0/8* (0.0%) | 3/8* (37.5%) |
| Cover slips | 8/40 (20.0%) | 5/15 (33.3%) | 3/25 (12.0%) | 0/8 (0.0%) | 0/8 (0.0%) | 8/8 (100.0%) |
| Disposable scalpel set | 0/39* (0.0%) | 0/14* (0.0%) | 0/25 (0.0%) | 0/0 (0.0%) | 0/0 (0.0%) | 0/0 (0.0%) |
| *Reagents* | | | | | | |
| Methylene Blue | 16/40 (40.0%) | 9/15 (60.0%) | 7/25 (28.0%) | 15/16 (93.8%) | 0/16 (0.0%) | 1/16 (6.3%) |
| Giemsa stain solution | 14/40 (35.0%) | 8/15 (53.3%) | 6/25 (24.0%) | 7/14 (50.0%) | 0/14 (0.0%) | 7/14 (50.0%) |
| TB Ziehl-Neelsen kit | 14/40 (25.0%) | 7/15 (46.7%) | 7/25 (28.0%) | 13/14 (92.9%) | 0/14 (0.0%) | 1/14 (7.1%) |
| Normal saline | 11/40 (27.5%) | 7/15 (46.7%) | 4/25 (16.0%) | 1/10* (10.0%) | 0/10* (0.0%) | 9/10* (90.0%) |
| Grouping sera | 9/40 (22.5%) | 5/15 (33.3%) | 4/25 (16.0%) | 0/9 (0.0%) | 0/9 (0.0%) | 9/9 (100.0%) |
| Sodium | 6/40 (15.0%) | 4/15 (26.7%) | 2/25 (8.0%) | 0/6 (0.0%) | 0/6 (0.0%) | 6/6 (100.0%) |
| Absolute ethanol | 4/40 (10.0%) | 1/15 (6.7%) | 3/25 (12.0%) | 2/4 (50.0%) | 0/4 (0.0%) | 2/4 (50.0%) |
| Distilled water | 3/40 (7.5%) | 3/15 (20.0%) | 0/25 (0.0%) | 1/3 (33.3%) | 0/3 (0.0%) | 2/3 (66.7%) |

Data is presented in absolute number of CHCs (percentage of sample).

*Missing data. NA = Supply sources for equipment were not collected for this study.

urine/stool containers (25.0%) led to recycling of old containers, often without having access to proper sterilisation methods (absolute ethanol, for example, was only available in only 10.0% of the CHCs). This makeshift laboratory work also revealed some of the ambiguities in the meaning of "availability'"- staff ensured a test remained *doable* despite the physical components it formally required not being *present*—even while it raised questions about the quality of laboratory tests conducted under such conditions.

## Human resources

The Basic Package states that CHC laboratories should be staffed by both a permanent laboratory technician and a laboratory assistant. Only 17.5% (*n/N* = 7/40) of the CHCs met this

criteria. 22.5% ($n/N$ = 9/40) of CHCs had a permanent laboratory technician. Of the 31 facilities where there was no permanent laboratory technician, 16.1% ($n/N$ = 5/31) were staffed by a volunteer laboratory technician. In 42.5% ($n/N$ = 17/40) of the CHCs there were no permanent or volunteer laboratory technicians nor permanent or volunteer laboratory assistants available, and lab tests were generally performed by other health workers including nurses or pharmacy assistants.

However, we also noted that several facilities, mostly those in the Urban district, had more laboratory staff than outlined in the Basic Package. For example, 4 facilities reported more than 1 permanent laboratory technician on staff. In terms of comparison between the two districts surveyed, human resources were for the most part more available in Rural District as opposed to Urban District. For example, a greater proportion of facilities in Rural District had a permanent lab technician on staff (26.7%, n/N = 4/15 in rural CHCs; versus 20.0%, n/N = 5/25 in urban CHCs).

Six out of the seventeen reported RDTs showed statistically significant association between availability and having a permanent lab technician as part of the CHC staff. Relatively strong effect sizes favoured test availability when a permanent lab technician was present (Table A in S1 Data). These included Haemoglobin RDT ($p$ = 0.008, Cramer's V = 0.448), Urinalysis multi-stick RDT ($p$ = 0.008, Cramer's V = 0.448), H pylori RDT ($p$ = 0.030, Cramer's V = 0.419), Random blood sugar RDT ($p$ = 0.021, Cramer's V = 0.393), Hepatitis B profile RDT ($p$ = 0.034, Cramer's V = 0.382) and Hepatitis B RDT ($p$ = 0.029, Cramer's V = 0.380).

Six out of the twelve reported manual assays showed statistically significant association between test availability and the presence of permanent lab technician. Both strong and moderate effect sizes favoured test availability in the presence of a permanent lab technician (see Table B in S1 Data). The tests with strong effect sizes included Blood typing ($p$ = 0.001, Cramer's V = 0.713), Widal test ($p$ = 0.001, Cramer's V = 0.572), Skin snip (p = 0.001, Cramer's V = 0.572), Malaria microscopy ($p$ = 0.007, Cramer's V = 0.473), while moderate effect sizes included Biological specimen routine analysis ($p$ = 0.044, Cramer's V = 0.358.), TB sputum microscopy ($p$ = 0.044, Cramer's V = 0.358.).

## Laboratory infrastructure

Table 2 shows the laboratory infrastructure and human resources available at the CHCs alongside the WHO laboratory standard that most closely aligns to the survey question [26] (see also Fig B in S1 Data). No single CHC met all of the WHO criteria for laboratory standards. 55% (n/N = 22/40) of CHCs had a designated laboratory space, but there was no clear division between those facilities that had a designated laboratory space and those that did not, with most facilities meeting some standards but not others. The results also show variability in the basic infrastructure and equipment needed to carry out diagnostic tests. Here, we highlight four key areas in which availability was low or varied substantially between facilities: water and electricity supplies, maintenance and repair systems, biosafety and waste management systems, and specimen transportation.

Access to clean running water was a challenge in the majority of CHCs. Out of the 40 CHCs surveyed, 35.0% ($n/N$ = 14/40) had a running tap and 57.5% ($n/N$ = 23/40) of facilities had a sink in the laboratory. When there was no running water, laboratory staff would fetch water from water tanks in other parts of the facility and might ask patients to buy water sachets from vendors to be used to rinse urine bottles. Furthermore, 70% of CHCs ($n/N$ = 28/40) had electricity in all parts of the facility but less than half of the CHCs in both districts had a working standby generator available (40.0%, $n/N$ = 6/25 in rural district, 44.0%, $n/N$ = 11/25 in

**Table 2. Availability of laboratory infrastructure and human resources.**

| WHO Laboratory Standards | DiaDev survey proxy laboratory standards | Total availability n/N (%) | Rural availability n/N (%) | Urban availability n/N (%) |
|---|---|---|---|---|
| Laboratory staff with appropriate qualifications and training | Laboratory technician–permanent staff | 9/40 (22.5%) | 4/15 (26.7%) | 5/25 (20.0%) |
| | Laboratory technician–permanent staff + volunteers | 14/40 (35.0%) | 8/15 (53.3%) | 6/25 (24.0%) |
| | Laboratory assistant–permanent staff | 14/40 (35.0%) | 3/15 (20.0%) | 11/25 (44.0%) |
| | Laboratory assistant–permanent staff + volunteers | 20/40 (50.0%) | 9/15 (60.0%) | 11/25 (44.0%) |
| | Laboratory technician (permanent staff) + laboratory assistant (permanent staff) | 7/40 (17.5%) | 2/15 (13.3%) | 5/25 (20.0%) |
| Adequate space for laboratory work | Designated lab space | 22/40 (55.0%) | 12/15 (80.0%) | 10/25 (40.0%) |
| Clean running water | Running tap at the facility | 14/40 (35.0%) | 5/15 (33.3%) | 9/25 (36.0%) |
| | Sink at the laboratory | 23/40 (57.5%) | 10/15 (66.7%) | 13/25 (52.0%) |
| | Water purification chemicals or filer | 7/40 (17.5%) | 4/15 (26.7%) | 3/25 (12.0%) |
| Lighting | Electricity in all parts of the facility | 28/40 (70.0%) | 8/15 (53.3%) | 20/25 (80.0%) |
| | Electricity in some parts of the facility | 1/40 (2.5%) | 1/15 (6.7%) | 0/15 (0.0%) |
| | Solely dependent on renewable energy | 10/40 (25.0%) | 6/15 (40.0%) | 4/25 (16.0%) |
| Back-up power | Standby facility available in working condition | 17/40 (42.5%) | 6/15 (40.0%) | 11/25 (44.0%) |
| Sanitation facilities | Sanitation facilities (patients & staff) | 39/40 (97.5%) | 15/15 (100.0%) | 24/25 (96.0%) |
| Drainage systems | Drainage system | 31/40 (77.5%) | 12/15 (80.0%) | 19/25 (76.0%) |
| Refrigerators and freezers | Refrigerator | 28/40 (70.0%) | 10/15 (66.7%) | 18/25 (72.0%) |
| Records of maintenance and repair procedures | Laboratory equipment is being serviced | 3/35* (8.6%) | 1/15 (6.7%) | 2/20* (10.0%) |
| | Maintenance record | 2/35* (5.7%) | 2/15 (13.3%) | 0/20* (0.0%) |
| SOPs for biosafety and handling samples | SOPs for performing tests | 10/40 (25.0%) | 5/15 (33.3%) | 5/25 (20.0%) |
| First aid kits | First aid kits | 0/40 (0.0%) | 0/15 (0.0%) | 0/25 (0.0%) |
| Appropriate Personal Protective Equipment (PPE) | Nitrile disposable gloves | 26/40 (65.0%) | 13/15 (86.7%) | 13/25 (52.0%) |
| | Laboratory coat | 9/40 (22.5%) | 5/15 (33.3%) | 4/25 (16.0%) |
| Container for sharps disposal | Container for sharps disposal | 40/40 (100.0%) | 15/15 (100.0%) | 25/25 (100.0%) |
| Separate waste disposal for infectious and non-infectious waste | Display of posters near waste bins to guide users to segregate waste | 27/40 (67.5%) | 6/15 (40.0%) | 21/25 (84.0%) |
| | Segregation of waste at point of final disposal | 21/38* (55.3%) | 8/14* (57.1%) | 8/14* (57.1%) |
| | Functional incinerator | 19/40 (47.5%) | 10/15 (66.7%) | 9/25 (36.0%) |
| | Burial pit | 13/40 (32.5) | 6/15 (40.0%) | 7/25 (28.0%) |
| Specimen transportation | Specimen transportation done | 14/40 (35.0%) | 2/15 (13.3%) | 12/25 (48.0%) |

Data is presented in absolute number of CHCs (percentage of sample).

*Missing data

urban district). Lack of fuel and mechanical problem reported as challenges underlying non-functioning generators.

In facilities where it was reported, three facilities (8.6%, n/N = 3/35) cited that their laboratory equipment was not being serviced at a regular interval but only "anytime a biomedical engineer is available". Only two facilities (5.7%, n/N = 2/35) had a maintenance record. In cases of breakdowns of equipment such as microscope, incubator, centrifuge, and glucometer, most facilities reported they would arrange repair themselves through sourcing a locally based non-specialised engineer. Some facilities reported writing a written request to the District Health Management Team to deal with the repair, and stated it can take over a year before it is worked on. Some facilities also reported not knowing what to do when there is a breakdown of equipment.

We found significant gaps in the provision of biosafety training, and systems and infrastructure for waste management. Only a quarter of the health facilities reported to receive

regular biosafety training, with several staff reporting to have received biosafety training during Ebola, but in many cases this was not updated since. All 40 CHCs had a sharps box supplied through the national Infection Prevention Control programme, established post-Ebola with support from key partner organisations WHO, the United States Centers for Disease Control and Prevention (CDC) and the United Nations International Children's Emergency Fund (UNICEF) and all but one had a dustbin in the laboratory. We found that 32.5% (*n*/N = 13/40) of all CHCs had a burial pit to bury sharps. However, burial pits were sometimes makeshift constructions and did not always comply with guidance on depth. Less than half (47.5%, *n*/N = 19/40) of the total CHCs had a functional incinerator. In many cases non-functioning incinerators were reported to be fairly new and to have been built with donor funding during or after the 2014–2016 Ebola outbreak. Staff reported challenges such as "no fuel to burn", "no roofing to protect from rain", "cracks in walls due to poor construction", and "waiting for handover from donors" as reasons for non-functioning incinerators.

Posters were displayed in 67.5% (*n*/N = 27/40) of CHCs to guide users to segregate waste at the point of disposal. However, segregation of waste at point of final disposal was only done in only 55.3% (*n*/N = 21/38) of CHCs. Moreover, in several CHCs health workers reported they informally arranged for waste (including both general and infectious waste) to be collected by keke (tricycle) drivers or people using wheelbarrows who transported it to public dumping sites or city drains leading to the sea, posing public health risks for people, including children, conducting recycling activities.

Finally, we found that only 35% (*n*/N = 14/40) of CHCs reported to have specimen transportation available at the facility to transport samples to higher-level facilities. In most cases, specimen transportation was done by a surveillance officer from the District Health Management Team who picked up samples for epidemic-prone diseases, such as measles. In one CHC, having no access to a microscope, the laboratory staff used public transport to personally transport blood samples to another CHC to conduct sample analysis in order to prevent the patient from travelling to a different facility. In general, however, CHCs referred the patients, and not the sample, to district or referral hospitals. This often led to diagnostic tests that were carried out at the CHC being repeated at higher levels of the health system, placing a higher burden on patients in terms of sample extraction and cost [22].

In terms of comparison between Urban and Rural Districts, laboratory space was available in a greater number of facilities in Rural district (80.0%, n/N = 12/15) versus Urban district (40.0%, n/N = 10/25). Some standalone infrastructures and items of equipment were also more available in Rural District than Urban District, but infrastructure that depends on large technical systems was for the most part more available in Urban District than Rural District. For example, running water was slightly more available in facilities in Urban District (36.0%, *n*/N = 9/25) than Rural District (33.3%, *n*/N = 5/15). Likewise, higher numbers of facilities in Urban District had electricity available in all parts of the facility (80%, *n*/N = 20/25) as compared to Rural District (53.3%, *n*/N = 8/15). In the Rural District a higher percentage of facilities (40.0%, *n*/N = 6/15) were dependent solely on off-grid energy sources—most commonly solar energy—in comparison to Urban District (16.0%, *n*/N = 4/25).

## Supply systems

The survey showed that supply chains were fragmented and unreliable, including a high dependence (>50%) on informal private sources for the majority of the available RDTs and manual assays. Fig 3 shows the supply of RDT kits by district. One RDT type, for Hepatitis B, was dependent entirely on private supply in both districts where it was available. In addition, when they were available in Rural District, Hepatitis C RDT, Syphillis RDT, Faecal Occult

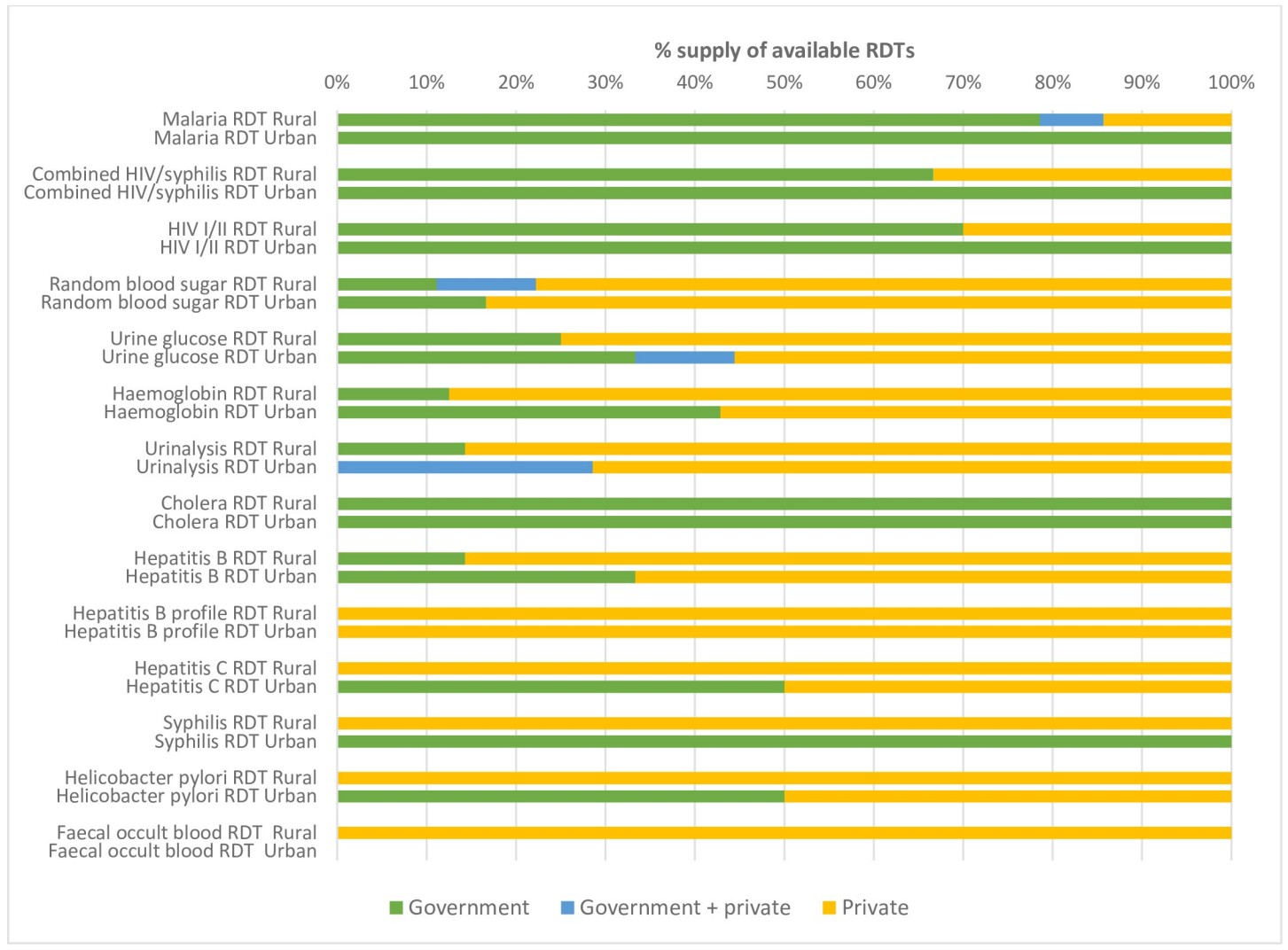

**Fig 3. Supply sources of available RDT.**

Blood RDT and Heliobacter Pylori were sourced entirely from a private supplier. Several other RDTs were highly dependent on private suppliers where they were available, including the commonly used multiparameter dipsticks used for urinalysis, or Haemoglobin RDT (including both filter paper tests and point-of-care haemoglobin meters), which were supplied via private sources in 78.6% ($n$/N = 11/14), and 73.3% ($n$/N = 11/15) of the CHCs where they were available respectively.

Since the supplies from the government-managed supply systems (particularly the free health care commodities) were often late or insufficient, in-charges reported making several trips per month, using their own money, to visit the District Medical Stores to request additional stock by filling in a paper-based request. In Western Area Urban District, cost-recovery commodities were arranged via the Freetown City Council, which procured supplies via private bidders. CHCs can make a request to the Mayor's office of the City Council for specific supplies, but reported that delays meant it was often quicker to purchase diagnostic supplies themselves from private sellers. In Western Area Rural District, there was no specific cost-recovery supply chain, and health workers reported receiving only free health care

commodities or disease-specific supplies. In interviews, health workers mentioned it was unclear which, and how many, diagnostics ought to be supplied under the free health care system, and which are cost-recovery. To supplement government supplies, CHCs procured additional tests and reagents through private supply chains, which they sourced via colleagues working and/or owning private laboratories, pharmacies, or through private sellers visiting facilities.

The Cholera RDT was the only RDT which was solely supplied by the government in all the CHCs where it was available, including in both districts. The Cholera RDT was also the only diagnostic that was supplied through the Central Public Health Reference Laboratory, located in Western Area Rural, directly to the DHMTs, and facility in-charges requested these tests by travelling to the DHMT and putting in a paper-based request. Other tests for which a large number of facilities relied exclusively on government supplies include those RDTs (Malaria RDT, HIV Syphilis RDT, HIV I/II RDT) that are managed via a Global Fund focal person housed at the DHMT. These were also the three tests that were most commonly available overall. But even for tests supplied by the Global Fund, there was variability in supply and some facility staff from CHCs in Western Area Rural District reported using their own money to travel to the DHMT to request additional malaria RDTs, or buying additional malaria RDTs from pharmacies or private sellers visiting their facilities. In three of the facilities where the Malaria RDT was available at the time of data collection, the tests were supplied by private sources. While the three most commonly available RDTs were supplied primarily via government, other commonly available tests (for example urine blood sugar and urine pregnancy) were primarily dependent on private supply where they were available.

We also asked facility staff whether the manual assays were supplied by government, private or a mixture of sources, as shown by Fig 4. Manual assays require multiple materials and equipment and it was not always clear exactly which items respondents were referring to in their response to this question about test supply, but our ethnographic research in one CHC suggests it was most likely related to the supply of reagents, as lab staff tended to conflate tests with reagents when talking about stock outs, for example stating: "no work; no reagent".

Three manual assay tests (TB Sputum Microscopy, Blood Microscopy, Fungal Identification) were exclusively supplied by government sources, while three others (Widal Test, Blood Typing, White Blood Cell Count) were exclusively supplied by private sources. In all of the nine CHCs where blood typing was done, the reagents used for this, grouping sera, were supplied via private sources, indicating the efforts lab staff undertook to make this available. Overall, the responses show that for manual assays, as for RDTs, the Rural district relied more on private sources in comparison to the Urban district where tests were available.

## Missing data

Data on the availability of syphilis RDT diagnostic test was missing from four CHCs, due to a data collection omission during the CHC visits. The CHCs with missing data for test availability were tested for differences in baseline characteristics from those included in the bivariate analyses compared to CHCs without missing data. No differences were detected.

Data on maintenance of laboratory equipment and availability of maintenance records was missing for five CHCs each, as the research assistant observed little to no lab equipment at the CHCs and therefore judged these questions to be non-applicable. Furthermore, four survey items for waste management had missing data due to the inability of the research assistant to observe these appropriately, which included the destruction of hypodermic needles (four CHCs), segregation of waste at point of disposal (two CHCs), destruction of syringe nozzles (six CHCs) and whether hypodermic needles were re-capped (five CHCs). From one CHC,

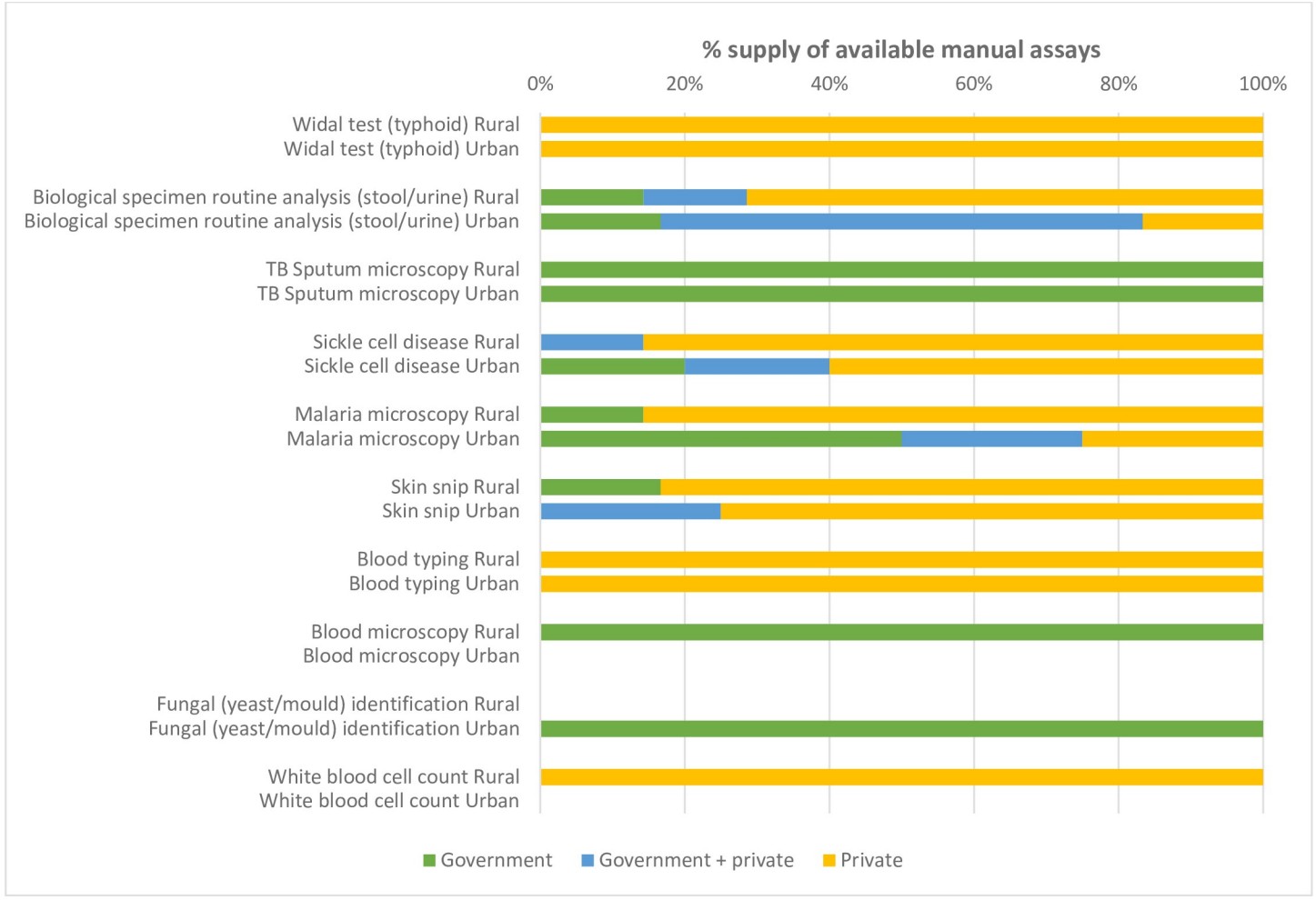

**Fig 4. Supply sources of available assay-based tests.** Note: Empty bars represent where there was no availability of tests.

data pertaining to supply sources of multi stick urine analysis test, syphilis and HIV duo test, normal saline reagent and needles and syringes were missing due to CHC staff being uncertain about supply sources.

## Discussion

Our results show little improvement in availability by comparison with the nation-wide assessment of laboratory services across primary and secondary facilities in all five regions of Sierra Leone that was conducted in 2015 [25]. This is despite the attention that the epidemic drew to the country's weak laboratory systems [26, 27], and the influx of international laboratory strengthening projects in its aftermath [28–30]. Western Area's good transportation infrastructure, its proximity to Freetown, and its links to tertiary referral hospitals and laboratory services, mean the availability of essential diagnostics in the area's CHCs are likely to be higher than for other areas and present a best-case scenario for the country. Yet we found overall availability of essential diagnostic tests, especially manual assays, to be low and to vary considerably. These results possibly reflect the focus of post-Ebola laboratory strengthening on strengthening molecular units in tertiary care laboratories in tertiary care facilities and

national reference laboratories, with little investment in improving diagnostic systems at a primary care level [12, 31].

In light of these chronic shortages in test availability in Sierra Leone, it is vital that we look beyond basic measures of physical presence and consider the factors that determine test supply and useability, that is whether or not they are "ready to hand". Our findings suggest three key areas for prioritisation: fragmented supply systems; limited human resources and dependency on volunteer labour; deficiencies in basic infrastructure and systems for maintenance, repair and waste management.

We found that the fragmentation of medical supply systems in Sierra Leone had a major impact on the testing landscape in Western Area. Our results show the high dependency of facilities on private sources for supply, which requires extra work by health facility staff to identify suppliers and negotiate prices, with additional costs usually being passed onto patients who are are being charged to recoup the costs [see also 21, 22]. It is especially concerning that those disease-specific RDT kits that were most commonly available were financed primarily through the Global Fund rather than through the free health care system or cost-recovery care system, suggesting that without additional support in supply and procurement, RDTs can be as vulnerable as reagents to weaknesses in government supply.

There was a greater dependence on private supply chains where tests were available in Rural district versus Urban. In the Urban district, a high number of facilities sourced some tests exclusively from government sources, including urine pregnancy test, syphilis RDT and helicobacter pylori, while a high number of rural facilities depended on private supplies for the same tests. The greater dependence on private supply chains in the Rural district was possibly related to the presence of the local governmental cost-recovery supply system in the Urban district, which lowers the need for health care workers to augment diagnostics supplies themselves. The greater distance, time and costs to travel to the District Medical Stores in Freetown for Rural CHCs than for Urban CHCs, might also explain why rural CHCs would opt for arranging supplies via private sources nearer to their workstation.

Very few CHCs met basic requirements for staffing. In particular, we found a shortage of permanent lab technicians across both districts. A statistically significant association between the availability of several tests and having a permanent laboratory technician on staff suggests that qualified laboratory staff may play a crucial role in bolstering the broken supply system. It is notable that this association also held true for some RDT kits, suggesting that the deployment of RDTs may not entirely obviate the need for experienced and skilled laboratory personnel [8, 32].

The dependence on laboratory assistants is especially concerning at this level of the health system since they often have no formal training in laboratory work. It is important to note, however, that a nationwide assessment carried out in 2015 found that in 30% of all 181 CHCs surveyed there was not a single laboratory worker [25]. By comparison, our survey pointed to a greater availability of laboratory assistants (50%) and laboratory technicians (35.0%), though when volunteer workers are excluded these percentages were much lower (35.0% and 22.5%). Analysis of underlying reasons for staffing shortages or surpluses was not part of our study, but possible explanations include the miscategorisation of facilities as CHCs within formal government documentation, and a moratorium on the recruitment of new staff by government institutions that was instituted in 2018.

The dependence on volunteers, who do not receive a regular salary and are not included in any official trainings, to provide testing services in Sierra Leone is also concerning. Qualitative findings from a study about volunteer nurses' experiences working in Ebola treatment units in Sierra Leone, showed that when promises of formal employment were not fulfilled, this resulted in feelings of institutional humiliation and low staff morale, which could possibly

impact on performance [33]. Recent insights from the COVID-19 response in the country suggest that memories of uncompensated risks during Ebola linger among front-line workers, some of whom were volunteers, making them unwilling "to risk their life again for COVID" [34]. Our study shows that volunteer and unpaid labourers fill staffing gaps in Sierra Leone not only during times of epidemic outbreaks, but also during "routine" times. The normalisation of volunteer staffing in the Sierra Leone health system deserves further attention in outbreak preparedness and laboratory strengthening policies.

We found widespread absence of basic laboratory infrastructure. All facilities were affected in some way by the lack of reliable electrification and/or running water, refrigeration or sanitation systems. A particularly conspicuous area of neglect in terms of laboratory infrastructure was that of waste management. Diagnostic work generates a range of infectious and non-infectious waste, from used RDT casettes, to blood collection devices, swabs, gloves, and waste biological materials/reagents. According to WHO guidance, this waste should be carefully segregated at source and infectious waste should be incinerated and/or disposed of in a burial pit [35]. But we found waste management systems and infrastructure to be weak across the board. In particular, systems for segregating waste were frequently lacking and many of the incinerators installed during or after Ebola were non-functional. The lack of waste management infrastructure generated hazardous working conditions for CHC staff and occupational scavengers at the municipal dumps where large quantities of diagnostic waste ended up, in addition to burning of waste in burial pits potentially contributing to damaging levels of air pollution in local communities. Diagnostic waste management is a fundamental aspect of diagnostic provision at the primary care level and needs to be addressed with a much greater sense of urgency. Indeed, we would question whether testing equipment that is deployed without basic waste management systems and infrastructure being put in place can truly be said to be available, in the sense of being "ready-to-hand" to laboratory staff.

As we reported above, no single facility surveyed for this study met all WHO laboratory standards [36]. Yet, in many cases where essential laboratory criteria were not met, some equipment, infrastructure and expertise was nonetheless available. We found that these gaps were often bridged by staff, who improvised with what was to hand to carry out manual tests that usually require the presence of a fully-equipped laboratory. This demonstrates that what counts as a laboratory in under-resourced settings can be ambiguous, with potentially important implications for the application of the WHO's EDL.

The EDL makes a distinction between tests that should be made available at facilities with and those without a laboratory. However, our findings suggest that a binary distinction between the absence/presence of a laboratory can be misleading, and that it is important to focus instead on whether particular tests are "doable" with the configuration of human resources, systems, equipment and infrastructure that are available. Moreover, the EDL does not prescribe what kind of facility and at what level of the health system *should* have a laboratory. The risk of such an approach is that it implies that when and where laboratory infrastructure and expertise are unavailable–for example because of lack of investment in laboratory staff and infrastructure—RDTs are an acceptable substitute, and therefore does little to incentivise investment in existing laboratory systems at the primary care level. A potential effect of the EDL's focus on *essential* diagnostics is therefore that it can lead to a race to the bottom in terms of what governments are willing to fund and support [37].

Rather than classifying tests by those that should be made available at facilities with/without a laboratory, as is done in the EDL, we suggest that it would be more effective to identify which tests should be available at each level of the health system—as is done by the Basic Package in Sierra Leone–and then establish the specific infrastructure, equipment, staff and systems that need to be put in place for those tests to be "ready-to-hand" for staff. This focus on

infrastructure and systems should ideally inform updates to the Basic Package and its development into a national EDL for Sierra Leone.

## Limitations

We did not include quality assurance systems in our survey because no external quality assessments systems were running in the districts that we surveyed at the time of data collection (quality assurance systems were provided at secondary level laboratories, and country-wide audits of laboratory quality were undertaken) [25]. Given the likely damaging impact of the infrastructural deficiencies that our study identified on test quality, support for quality assurance systems at the primary care level clearly needs to be a funding and policy priority at national and international levels.

We only assessed diagnostic availability in one administrative sub-division, consisting of two small adjoining districts, albeit the most populous sub-division in the country. These districts were close to the central medical distribution stores, hence it is likely that supply problems which exist in these areas might be even worse in areas further away. On the other hand, a higher concentration of private suppliers of laboratory equipment in and around Freetown likely impacted the survey results for supply source. The supply analysis involved fairly crude differentiation of public and private sources. Our findings point to the need for a more nuanced and detailed analysis of supply systems in the region in future research. Since we only checked the supply source for tests which were available, we are unable to provide any associations between the supply source and availability of specific tests.

In terms of the comparability of our survey to the EDL, a limitation is that nine tests that were included in the version of the EDL available at the time of data collection [38] were not assessed since we followed the Sierra Leone Basic Package document as reference.

## Conclusion

Despite growing awareness of the importance of access to diagnostics at a primary care level in Low and Middle Income Countries, and notwithstanding substantial investments in laboratory strengthening in Sierra Leone during and after the West Africa Ebola Outbreak, availability of diagnostic tests in Sierra Leone's Community Health Centres remains markedly low. Lower levels of availability for manual assays, and routine shortages in the equipment and reagents on which these tests depend, illustrate the particular challenges in extending laboratory infrastructure to a primary care level. Yet we also found that health systems and infrastructure issues impacted the supply and useability of RDTs.

Three issues that affect the availability and useability of RDTs as well as manual assays in Sierra Leone standout from the analysis. First, RDTs are shown to be susceptible to the same supply pressures as manual assays. Most strikingly, in the context of Sierra Leone's fragmented supply system, the ready supply of RDTs is shown to depend on the presence of qualified laboratory technicians at health facilities; this is despite the tests themselves being designed for use by non-experts. Second, the existing waste management infrastructure at a primary care level is inadequate for processing a growing influx of disposable, single-use RDT devices. Third, the non-existence of routine quality assurance systems threatens the quality of RDT tests and potentially undermines the trust of staff and patients in the results. While the development of affordable, easy-to-use RDTs addresses some of the challenges involved in extending diagnostic systems and infrastructure to primary care settings, for these tests to be truly "ready to hand" for health facility staff, qualified personnel, supply systems, waste management systems and quality assurance systems still need to be in place.

The issue of useability and the complex assemblage of factors involved in making a test "ready to hand" for health facility staff distinguishes diagnostic tests from essential medicines, which often come in an easily administered pill-based format. While the launch of the WHO's EDL marks a major milestone on the road to universal access for diagnostics, there is also scope for the EDL to be expanded beyond its current narrow focus on the physical availability of diagnostic tests and to include recommendations and guidance regarding the human resources, infrastructure, and systems upon which the capacity to *do* tests depends.

Our focus on what makes tests doable also highlights the need for national-level investment in basic systems and infrastructure as a basis for improving diagnostic availability at a primary care level. This is important both for routine primary care and for national surveillance systems, since first cases in epidemics are most likely to present at peripheral health facilities. Our results indicate several priority areas where a holistic approach to diagnostic availability could make a difference, including the harmonisation of diagnostic supply systems across individual disease and funding programmes (with public-private partnerships one possible means of achieving this), investment in healthcare waste management systems to reflect the growing use of single-use diagnostic devices in primary care, and greater investment in qualified laboratory technicians as the core foundation for an effective and accessible laboratory network.

## Supporting information

**S1 Data.**
(DOCX)

**S1 Text. Checklist for assessing diagnostics available at community health centres.**
(DOCX)

**S2 Text. Inclusivity in global research.**
(DOCX)

## Acknowledgments

We wish to thank the community health centres' management and staff, especially the laboratory workers, for sharing their insigths and time with us. We also wish to thank the Sierra Leone Ministry of Health and Sanitation for their support of the research project, and the Western Area Urban and Rural District Medical Officers and laboratory technicians for their assistance with facilitating introductions to the health centres. Furthermore we would like to acknowledge the Laboratory Technical Working Group, in particular Dr Isatta Wurie, for providing her technical guidance in the early stages of the project. We also wish to thank Ifunanya Nnaemeka, Claire Nankoma, Thaimu Bangura, Sorie Samura and Fenella Beynon from King's Sierra Leone Partnership for their input during the early stages of the survey tool development, as well as Dr Ann Kelly for her intellectual guidance throughout the project. We are grateful to Roxanne Connolly for her feedback on an early draft and advice on the quantitative methods employed for this paper. Finally, we thank Dr Madhukar Pai for his methodological advice and sharing of a checklist used to assess diagnostic availability, which was highly valuable for the development of our survey tool.

## Author Contributions

**Conceptualization:** Alice Street, Eva Vernooij, Rashid Ansumana.

**Data curation:** Alice Street, Eva Vernooij, Mats Stage Baxter.

**Formal analysis:** Mats Stage Baxter.

**Funding acquisition:** Alice Street.

**Investigation:** Alice Street, Eva Vernooij, Francess Koker, Fatmata Bah.

**Methodology:** Alice Street, Eva Vernooij, Francess Koker, Mats Stage Baxter, Fatmata Bah, Momoh Gbetuwa, Mikashmi Kohli, Rashid Ansumana.

**Project administration:** Alice Street, Eva Vernooij.

**Resources:** Alice Street.

**Supervision:** Alice Street, Eva Vernooij.

**Validation:** Alice Street, Eva Vernooij.

**Writing – original draft:** Alice Street, Eva Vernooij, Mats Stage Baxter, Rashid Ansumana.

**Writing – review & editing:** Alice Street, Eva Vernooij, Francess Koker, Mats Stage Baxter, Fatmata Bah, James Rogers, Momoh Gbetuwa, Mikashmi Kohli, Rashid Ansumana.

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
