## [Decision Letter · Decision Letter 0]

2 Jun 2022

PGPH-D-22-00241

The “ready-to-hand” test: Diagnostic availability and usability in primary health care settings in Sierra Leone

Dear Dr. Street,

Thank you for submitting your manuscript to PLOS Global Public Health. After careful consideration, we feel that it certainly has merit but the main text could be organised in a more concise and effective manner, as suggested by Reviewer 2. Therefore, we invite you to submit a revised version of the manuscript that addresses the points raised during the review process.

Please submit your revised manuscript by . If you will need more time than this to complete your revisions, please reply to this message or contact the journal office at globalpubhealth@plos.org. Please include the following items when submitting your revised manuscript:

We look forward to receiving your revised manuscript.

Kind regards,

Marco Liverani, PhD, MSc

Academic Editor

Journal Requirements:

2. Please update your Competing Interests statement. If you have no competing interests to declare, please state: “The authors have declared that no competing interests exist.”

3. Please provide separate figure files in .tif or .eps format only and ensure that all files are under our size limit of 10MB.

Additional Editor Comments (if provided):

Reviewers' comments:

Reviewer's Responses to Questions

**Comments to the Author**

1. Does this manuscript meet PLOS Global Public Health’s publication criteria? Is the manuscript technically sound, and do the data support the conclusions? The manuscript must describe methodologically and ethically rigorous research with conclusions that are appropriately drawn based on the data presented.

Reviewer #1: Yes

Reviewer #2: Yes

2. Has the statistical analysis been performed appropriately and rigorously?

Reviewer #1: I don't know

Reviewer #2: Yes

3. Have the authors made all data underlying the findings in their manuscript fully available (please refer to the Data Availability Statement at the start of the manuscript PDF file)?

Reviewer #1: Yes

Reviewer #2: Yes

4. Is the manuscript presented in an intelligible fashion and written in standard English?

Reviewer #1: Yes

Reviewer #2: Yes

5. Review Comments to the Author

Reviewer #1: This is a very comprehensive assessment of the availability of RDTs, laboratory infrastructure, and laboratory personnel at CHC in Sierra Leone. Well done.

Was the RDT and other equipment inventory physically viewed or counted during the interviews or based only upon verbal / written responses from the interviewee? Please add additional clarification in the Methods section.

Also, minor edit in Table 2 - "Lightning" should be "Lighting"?

Reviewer #2: One of the criteria for publication is that ‘PLOS Global Public Health does not copyedit accepted manuscripts, so the language in submitted articles must be clear, correct, and unambiguous’. It took me some time to click the YES button as part of my review. I do NOT mean to say the writers do not use English grammar properly, I DO mean to say that I think the paper can be significantly improved for the reader by doing a rewrite focused on telling a larger story. Currently the paper is too long (especially the introduction) and is a drip drip drip litany of all the things that are problematic with this or that rural or urban lab. The reader leaves with the sense that things are not well in Sierra Leone, but not with the larger story e.g. The presence of a trained laboratory scientist improves lab quality OR Rural labs in this province of Western Sierra Leone have a higher dependence on the private sector.

Detailed feedback.

1. Title – The meaning of the phrase ‘Ready to Hand’ is not obvious until it is defined also it does not communicate across different types of English speakers. Consider using just “Diagnostic availability and usability in primary health care settings in Sierra Leone” as the title.

2. Introduction

a. Too long

b. The first two paragraphs do not actually introcude the relevant subject to the reader. Rather it is a trope of dependency and ‘lack’ that many of us in the Global Health world – including me- have come to depend on. Resist the temptation. Delete these two paragraphs. It will bring the reader into the relevant material faster. Start with ‘Since the publication of the EDL…

3. Methods – Good

4. Results

a. Litany of problems, larger messages / themes were not clear. Yes I know this is the result section, not the discussion but if there is a larger thematic approach it will affect what gets highlighted.

b. In the figures, consider using other none-colorimetric ways of differentiating your various groups. Ways such as cross-hatching or dots. The advantage of this is that they still communicate even if one doesn’t have a color printer.

c. The percentage with tap water was 59%. Why was the denominator 39 instead of 40?

d. Consider expanding the paragraph that starts ‘ There was a greater dependency… ‘. Also consider putting it in the Discussion. I think this finding is important and will be of wide interest to your Global Health readership

5. Discussion

a. One gap was ..when items are sourced privately, how do the lab staff recoup their money? Same question for when they pay for transporting specimens. Are the costs being transferred to the patients?

6. PLOS authors have the option to publish the peer review history of their article (what does this mean?). If published, this will include your full peer review and any attached files.

**Do you want your identity to be public for this peer review?** For information about this choice, including consent withdrawal, please see our Privacy Policy.

Reviewer #1: No

Reviewer #2: No

---

## [Decision Letter · Decision Letter 1]

22 Nov 2022

PGPH-D-22-00241R1

The “ready-to-hand” test: Diagnostic availability and usability in primary health care settings in Sierra Leone

Dear Dr. Street,

Thank you for submitting your revised manuscript to PLOS Global Public Health. The manuscript is in good shape. However, prior to publication, we invite you to address the minor comments in the attached file .

We look forward to receiving your revised manuscript.

Kind regards,

Marco Liverani, PhD, MSc

Academic Editor

Journal Requirements:

Additional Editor Comments (if provided):

Please address minor comments in the attached document.

Reviewers' comments:

Reviewer's Responses to Questions

**Comments to the Author**

1. If the authors have adequately addressed your comments raised in a previous round of review and you feel that this manuscript is now acceptable for publication, you may indicate that here to bypass the “Comments to the Author” section, enter your conflict of interest statement in the “Confidential to Editor” section, and submit your "Accept" recommendation.

Reviewer #1: All comments have been addressed

2. Does this manuscript meet PLOS Global Public Health’s publication criteria? Is the manuscript technically sound, and do the data support the conclusions? The manuscript must describe methodologically and ethically rigorous research with conclusions that are appropriately drawn based on the data presented.

Reviewer #1: Yes

3. Has the statistical analysis been performed appropriately and rigorously?

Reviewer #1: I don't know

4. Have the authors made all data underlying the findings in their manuscript fully available (please refer to the Data Availability Statement at the start of the manuscript PDF file)?

Reviewer #1: Yes

5. Is the manuscript presented in an intelligible fashion and written in standard English?

Reviewer #1: Yes

6. Review Comments to the Author

Reviewer #1: Thank you for this resubmission. My comments from the last submission have been addressed.

7. PLOS authors have the option to publish the peer review history of their article (what does this mean?). If published, this will include your full peer review and any attached files.

**Do you want your identity to be public for this peer review?** For information about this choice, including consent withdrawal, please see our Privacy Policy.

Reviewer #1: No

---

## [Editor Report · Decision Letter 2]

14 Dec 2022

The “ready-to-hand” test: Diagnostic availability and usability in primary health care settings in Sierra Leone

PGPH-D-22-00241R2

Dear Dr Street,

We are pleased to inform you that your manuscript 'The “ready-to-hand” test: Diagnostic availability and usability in primary health care settings in Sierra Leone' has been provisionally accepted for publication in PLOS Global Public Health.

Best regards,

Marco Liverani, PhD, MSc

Academic Editor